# Reciprocal and dynamic polarization of planar cell polarity core components and myosin

**Erin Newman-Smith[1][†], Matthew J Kourakis[1][†], Wendy Reeves[1,2], Michael Veeman[1,2], William C Smith[1]\***

[1]Department of Molecular, Cell and Developmental Biology, University of California, Santa Barbara, Santa Barbara, United States; [2]Division of Biology, Kansas State University, Manhattan, United States

**Abstract** The *Ciona* notochord displays planar cell polarity (PCP), with anterior localization of Prickle (Pk) and Strabismus (Stbm). We report that a myosin is polarized anteriorly in these cells and strongly colocalizes with Stbm. Disruption of the actin/myosin machinery with cytochalasin or blebbistatin disrupts polarization of Pk and Stbm, but not of myosin complexes, suggesting a PCP-independent aspect of myosin localization. Wash out of cytochalasin restored Pk polarization, but not if done in the presence of blebbistatin, suggesting an active role for myosin in core PCP protein localization. On the other hand, in the *pk* mutant line, *aimless*, myosin polarization is disrupted in approximately one third of the cells, indicating a reciprocal action of core PCP signaling on myosin localization. Our results indicate a complex relationship between the actomyosin cytoskeleton and core PCP components in which myosin is not simply a downstream target of PCP signaling, but also required for PCP protein localization.

**\*For correspondence:** w_smith@lifesci.ucsb.edu

[†]These authors contributed equally to this work

**Competing interests:** The authors declare that no competing interests exist.

## Introduction

Despite the importance of the planar cell polarity (PCP) pathway in ensuring the proper orientation of cells in a range of embryonic and adult tissues, the molecular mechanisms of this pathway are not fully understood. The PCP pathway is ancient in metazoans, and its requirement for proper development and physiology has been well described in insects (e.g., *Drosophila*) and chordates. The function of the PCP pathway can be seen in the coordinated polarization of subcellular components such as cilia and multi-celled structures such as bristles in epithelial sheets and organs like the *Drosophila* wing, vertebrate inner ear, and kidney, as well as in dynamic processes, such as cell migration, convergence and extension of mesoderm, neural tube closure, and axonal guidance (*Ybot-Gonzalez et al., 2007*; *Wallingford, 2012*; *Tissir and Goffinet, 2013*; *Papakrivopoulou et al., 2014*). One of the complicating issues of PCP signaling in *Drosophila*, where it has been most extensively addressed, is that it appears to consist of several semi-independent pathways that may act either in parallel or in sequence (*Gray et al., 2011*; *Lawrence and Casal, 2013*). The best characterized of these pathways is the core PCP pathway, which acts locally to coordinate polarity between neighboring cells. The key components of the core pathway include the transmembrane proteins Strabismus/Van Gogh and Frizzled, and the cytoplasmic proteins Prickle and Dishevelled. Some components of this pathway are shared with the wnt/β-catenin signaling pathway, although unlike the Wnt/β-catenin pathway the core PCP pathway is thought to act primarily by directing cytoskeletal organization, rather than by regulating transcription.

While the core PCP pathway coordinates local polarity in groups of cells, there appears to be a requirement for an additional signaling pathway(s) that brings this local coordination into register

**eLife digest** Animal cells that form flat layers of a tissue, such as the skin or the lining of internal cavities, are often orientated in the same direction. The same is true for structures such as hairs or feathers, which are attached to the skin. This phenomenon is known as 'planar cell polarity' (or 'PCP' for short).

Many different organisms use similar mechanisms to establish this kind of tissue pattern. The best-studied mechanism involves the so-called 'core PCP pathway'. Signaling proteins in this pathway coordinate the polarity of neighboring cells. Other 'global signaling pathways' are thought to first ensure that tissues are correctly orientated within the embryo as a whole, and to do this, the global pathways are thought to align a network of filament-like structures within the cells in a particular direction. Once correctly orientated, these filaments—known as microtubules—have been proposed to help position the components of the core PCP pathway such that they can correctly orientate the rest of the cell.

Now Newman-Smith, Kourakis et al. have identified another network of filaments within cells that interacts with components of the core PCP pathway in a sea squirt called *Ciona savignyi*. This organism begins life as a tadpole-like larva that has a flexible rod-shaped structure, called a 'notochord', running along the length of its body. The cells of the notochord become polarized as they develop. When microtubules are disrupted, their planar polarity remains unaffected. However, when another network of filaments—called the actomyosin network—is chemically disrupted, the polarity of certain core PCP components is lost.

The findings of Newman-Smith, Kourakis et al. reveal that the core PCP components and the actomyosin network in this sea squirt reinforce each other's polarity. This represents an alternative to the previously described models of planar polarity in which the core PCP components are thought to drive the polarization of the actomyosin network. Whether this model extends to planar cell polarity mechanisms in other organisms, such humans and other animals with backbones, remains a question for future work.

---

with the axes of the organ or embryo. The precise nature of this so-called 'global' signaling pathway has been elusive, although several candidate pathways are supported experimentally. One model for the *Drosophila* global polarizer stresses the importance of mechanical forces generated during morphogenesis leading to the alignment of cells (*Eaton and Jülicher, 2011*). Two signaling pathways have also been proposed to act as global polarizers—the Wg/Wnt4 pathway (*Wu et al., 2013*), and the Dachsous (Ds)/Fat/Four-jointed (Fj) pathway (*Casal et al., 2006*; *Thomas and Strutt, 2012*; *Matis et al., 2014*; *Olofsson et al., 2014*). The Ds/Fat/Fj module has been most extensively characterized and appears to signal via formation of heterodimers with the extracellular domains of Ds and Fat (which are atypical cadherins), putatively in a gradient across the tissue (*Ambegaonkar et al., 2012*; *Bosveld et al., 2012*; *Brittle et al., 2012*). Most recently, it was reported that the Ds/Fat/Fj module affects polarity through microtubule orientation, which in turn directs core PCP polarization (*Matis et al., 2014*). Fat and Ds orthologs also play roles in vertebrate development, but their precise roles in regulating aspects of PCP remain to be clarified (*Mao et al., 2011*; *Saburi et al., 2012*). One final complicating factor in assessing the role of the Fat/Ds/Fj pathway in planar polarity is the apparent overlap with the growth-stimulating Hippo pathway (*Lawrence and Casal, 2013*).

The notochord of the ascidian *Ciona* provides a particularly tractable model for the study of the PCP pathway in tissue morphogenesis (*Kourakis et al., 2014*). Ascidians are invertebrate chordates, and as members of the chordate subphylum Tunicata they belong to the group of animals that are the closest extant relatives of the vertebrates (*Delsuc et al., 2006*). While the presence of the notochord—a stiff axial rod of mesodermal cells lying under the nerve cord—is a uniting feature of the Chordata, tunicate notochords are much simpler than those of vertebrates, and in *Ciona* the fully formed notochord consists of only 40 cells arranged in a stack one-cell wide (*Jiang et al., 2005*; *Kourakis et al., 2014*). We have described two discrete developmental phases in notochord morphogenesis that shows polarized cell behavior. Initially, the notochord precursor cells undergo a mediolaterally oriented intercalation behavior, which forms the notochord column along the AP axis. The role of the core PCP pathway in the convergent extension of the *Ciona* notochord is seen in the *pk*

mutant *aimless (aim)* of *Ciona savignyi*, which has defects in notochord cell intercalation behavior, morphology, and extracellular matrix deposition (*Jiang et al., 2005*; *Veeman et al., 2008*).

Following the completion of intercalation, the individual notochord cells initially assume a flattened-disk shape (*Jiang and Smith, 2007*). Over the next approximately 3–4 hr (corresponding to stages 21–24 [*Hotta et al., 2007*]), the cells undergo a dramatic elongation in the A/P axis, which results in the continued elongation of the entire tail. During the process of elongation, a second polarity becomes evident in the A/P axis (at a right angle to the mediolateral polarity seen earlier at intercalation). The first sign of A/P polarity can be seen shortly after the completion of intercalation (stage 22) in the localization of Pk protein to the anterior poles of the cells (*Kourakis et al., 2014*). As elongation continues the core PCP protein Strabismus (Stbm) also localizes to the anterior pole, while nuclei move to the posterior poles of the cells where they take on a flattened shape (*Figure 1*) (*Jiang et al., 2005*; *Kourakis et al., 2014*). The exception to this pattern is the posterior-most notochord cell whose nucleus is usually found in reverse orientation relative to the other cells (*Jiang et al., 2005*; *Kourakis et al., 2014*). In *aim* embryos, the nuclei of intercalated cells are still polarized, but they are randomly oriented to either the anterior or posterior pole of each cell (*Kourakis et al., 2014*). Following elongation, the notochord enters a new phase in its development (from stage 24 onward). A matrix is secreted into extracellular pockets that form between A and P faces of the cells (*Dong et al., 2009*; *Denker and Jiang, 2012*; *Deng et al., 2013*). As this process continues, the pockets of matrix between the cells expand and then fuse to make a single, uninterrupted lumen along the length of the notochord.

In this manuscript, we examine the relationship between core PCP signaling and the actin/myosin network. We report here that the notochord cells have anteriorly polarized myosin machinery. While existing models depict the polarization of myosin as downstream of PCP signaling, we instead present evidence for a more complex series of interactions in which the core PCP components and the myosin machinery act in a reciprocal fashion to promote cell polarization.

## Results

### Myosin is polarized in notochord cells

The *Ciona intestinalis* genome is predicted to contain six members of the non-muscle myosin II family (*Chiba et al., 2003*). One of these, myosin10/11/14/9, is reported to be expressed exclusively in the notochord (*Satou et al., 2001*) and is found in a two-gene operon with a myosin regulatory light chain (MRLC) (*Satou et al., 2008*). This MRLC is one of five found in the *C. intestinalis* genome and has previously been used to mark myosin assemblies in notochord cells (*Dong et al., 2011*). By convention, *Ciona* genes are named according to their vertebrate orthologs (*Stolfi et al., 2014*). This particular *Ciona* MRLC shares orthology to vertebrate MRLCs 9 and 12 (phylogeny not shown) and will be referred to here as MRLC9/12.

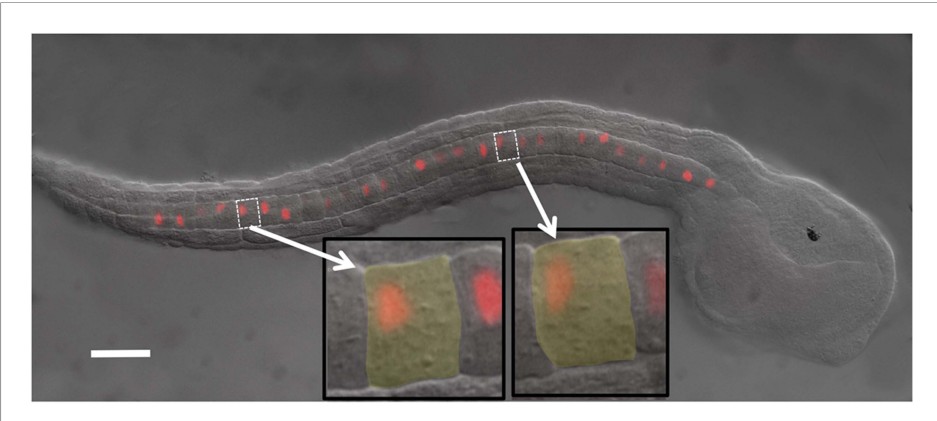

**Figure 1**. *Ciona intestinalis* late-tailbud embryo (stage 23) expressing an electroporated Histone 2A/Red Fluorescent Protein (H2A-RFP) in the notochord. Insets show two cells to illustrate the polarization of the nuclei to the posterior of the cells. Scale bar is 50 μm.

To visualize myosin assemblies in developing notochord cells, we constructed fusion proteins of MRLC9/12 and Myosin10/11/14/9 with Venus fluorescent protein and myc antigen, respectively. The resulting cDNAs were fused to the *Brachyury* cis-regulatory region to drive notochord expression (*Corbo et al., 1997*), and then introduced to embryos by electroporation. Low doses of plasmid were used to encourage highly mosaic expression. This allows the visualization of the exogenous proteins in cells isolated from the expression of neighboring cells, which would otherwise make the determination of anterior vs posterior protein localization difficult (*Jiang et al., 2005*; *Kourakis et al., 2014*). At the earliest developmental stage analyzed, stage 19—when the notochord cells are mid-intercalation, both Myosin10/11/14/9-myc and MRLC9/12-Venus were observed throughout the cells, with the highest localization on the perinotochordal surfaces where cells contact the basement membrane that surrounds the developing notochord (white arrows, stage 19; *Figure 2A*; see also *Veeman et al., 2008*). At the immediate completion of intercalation (stage 21, *Figure 2B*), the localization of both fusion proteins was still the strongest at the basal/lateral edges. However, less than an hour later, at stage 22, both proteins were found localized to the anterior pole of each notochord cell (arrowhead, *Figure 2C*), as well as continuing to be found at the basal/lateral edges. Quantification of MRLC9/12-Venus fluorescence signal along the anterior/posterior axis of notochord cells confirms a distinct transition in MRLC9/12-Venus localization between stages 21 and 22 (insets, *Figure 2B,C*). During notochord cell elongation, and at its completion (stage 24), the anterior localization of both proteins persists (*Figure 2D*). The onset of anterior/posterior localization of myosin assemblies closely matches the localization of Pk-myc, with A/P localization of Pk-myc starting at stage 22 (*Kourakis et al., 2014*). However, the Pk-myc and MRLC9/12 proteins do not appear to completely colocalize spatially. At stage 22, Pk-myc, Myosin10/11/14/9-myc, and MRLC9/12-Venus are each found to be enriched on the anterior side of the cells. However, as elongation progresses, Pk-myc localization becomes restricted to a small region within the center of this domain (*Jiang et al., 2005*; *Kourakis et al., 2014*), while Myosin10/11/14/9-myc and MRLC9/12-Venus remain more dispersed (*Figure 2*).

## Colocalization of phospho-MRLC with core PCP components

To examine the localization of active endogenous myosin complexes in the notochord, embryos were stained with an antibody to mono-phosphorylated (ser19) MRLC (pMRLC) (*Ikebe and Hartshorne, 1985*). Specific staining was observed along the cortices of the cells (white arrowheads, *Figure 3A*). Because the endogenous protein is not mosaically expressed, it is not possible to definitively resolve the expression to either the anterior or posterior face of the cells. Co-staining for electroporated Pk-myc and pMRLC shows relatively little overlap between the two proteins, with Pk-myc concentrated to a smaller domain at the center of the cell, as described above, while pMRLC distributed along the face of the cell. In contrast, a much higher degree of colocalization was observed for pMRLC and Stbm-Venus (*Figure 3B*, yellow arrowheads). As was reported previously (*Jiang et al., 2005*), Stbm is distributed evenly along the anterior face of the cell, rather than being concentrated centrally like Pk.

## Core PCP protein localization is lost upon disruption of the actin/myosin cytoskeleton

To examine possible relationships between polarization of the myosin machinery, core PCP (Pk-myc and Stbm-Venus) and nuclear polarity, embryos were treated with chemical inhibitors that disrupt the actin cytoskeleton (cytochalasin B and latrunculin) or myosin II motor activity (blebbistatin). Treatments were performed both to embryos with fully polarized notochord cells (stage 23), as well as to younger embryos (stage 21) before polarity is evident.

At stage 23, cytochalasin B caused notochord nuclei to detach from posterior membranes and drift anteriorly toward the middle of the cell (*Figure 4B,E*; and *Video 1*). In these same cells, we observed a loss of polarization of both Pk-myc (*Figure 4A,B*) and Stbm-Venus (*Figure 4D,E*). Embryos were scored for cytochalasin B treatment at both mid-stage 23 and late-stage 23, and only 6% (n = 35) and 5% (n = 20) of cells showed Pk-myc polarization, respectively. Even in cells scored as having polarized Pk-myc, the tight localization seen at the center of the anterior membrane was lost, replaced by labeling throughout the anterior membrane. By contrast, cells from embryos treated with DMSO at mid-stage 23 and late-stage 23 had 81% (n = 59) and 100% (n = 5) of cells with Pk-myc properly polarized, respectively. Similar results were seen for Stbm-Venus localization. In embryos treated with cytochalasin B at stage 23, only 6% (n = 50) had detectable Stbm-Venus localization vs 75% (n = 52) of

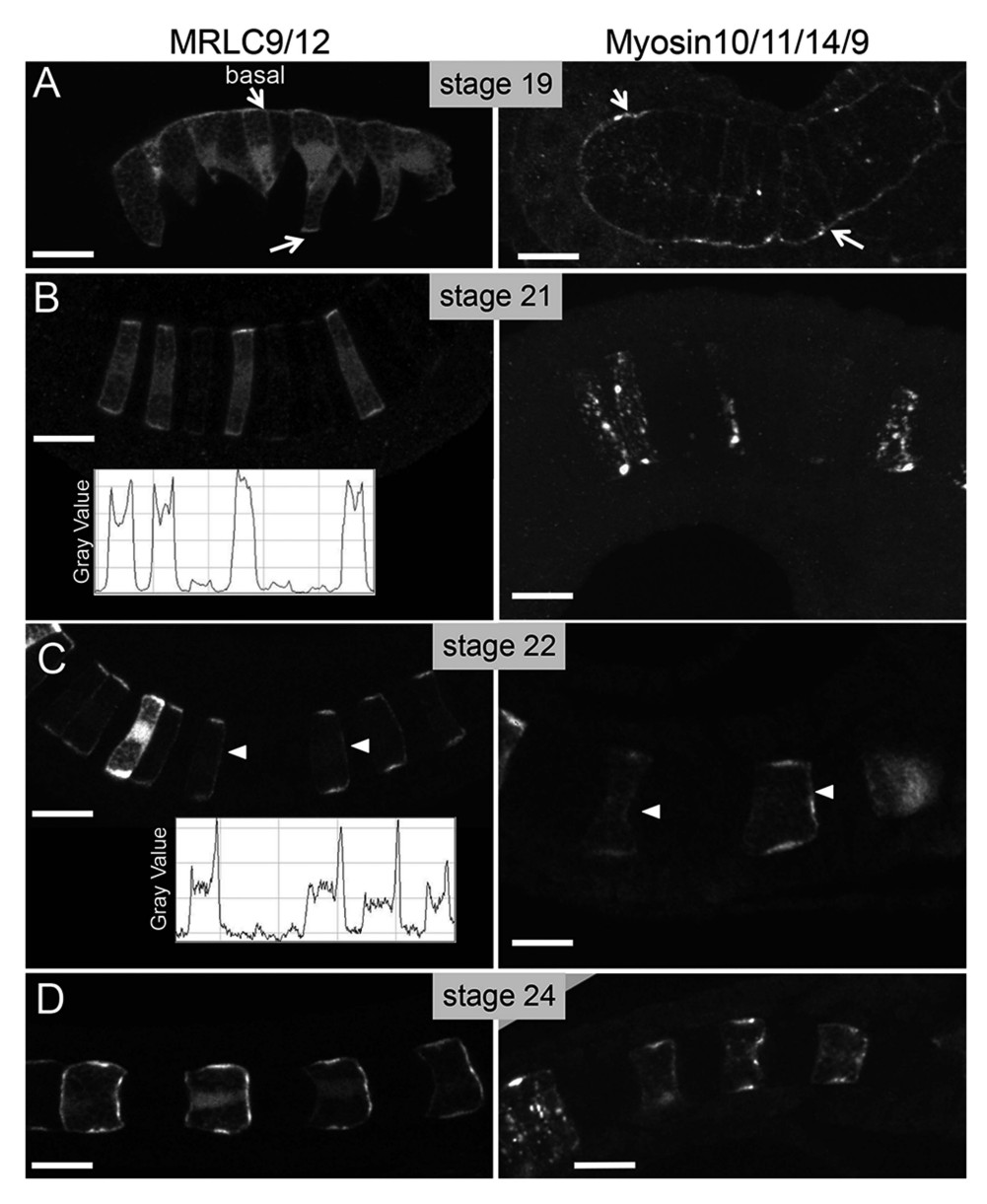

**Figure 2**. Time course for the subcellular localization of myc-tagged myosin10/11/14/9-tagged myosin regulatory light chain (MRLC) and Venus-tagged myosin regulatory light chain (MRLC9/12-Venus) at the stages indicated. Arrows in **A** indicate contacts of cells with the basement membrane, and arrowheads in **C** indicate anteriorly localized protein. Densitometry readings across the center of four cells demonstrate the transition in MRLC9/12 localization from stage 21 to stage 22 (charts in **B** and **C**). (**D**) At stage 24, the anterior localization of both proteins persists. Anterior is to the right in all panels. Scale bars are 10 μm.

control DMSO-treated cells. Latrunculin B, which inhibits actin polymerization by a different mechanism than cytochalasin B (*Morton et al., 2000*), also caused a loss of nuclear, Stbm-Venus, and Pk-myc polarization (*Figure 4C,F*). Only 5% and 4% of latrunculin B treated embryos showed localized Stbm-Venus and Pk-myc, respectively, vs 92% of controls (n = 57, 25, and 25). The stage 23 treatments demonstrated that cytochalasin B could disrupt the notochord cell polarity once it was established. To examine the effect of cytochalasin B on the establishment of polarity, embryos were treated at stage 21 and scored at stage 23. In these embryos, no polarization of Pk-myc was evident in any of the cells scored (n = 5) vs 92% of cells from vehicle (DMSO)-treated embryos (n = 12) having normal anterior polarization of Pk-myc.

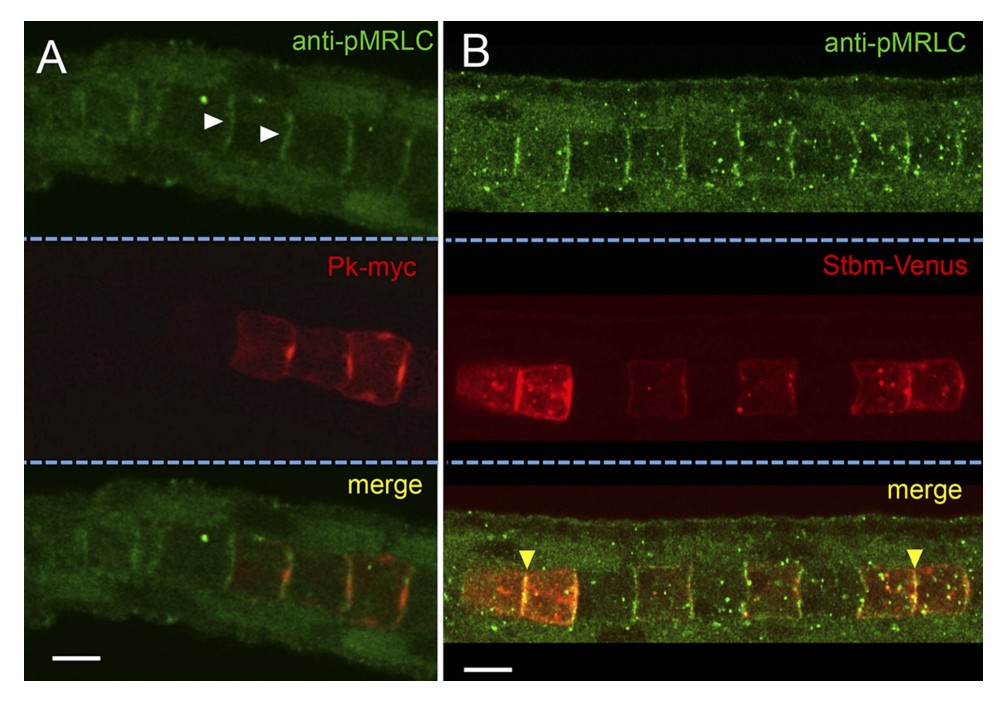

**Figure 3**. Subcellular localization of endogenous phospho-Myosin Regulatory Light Chain (pMRLC) and electroporated myc-tagged Prickle (Pk-myc) (**A**), and Venus-tagged strabismus (Stbm-Venus) (**B**). White arrowheads in **A** indicate anti-pMRLC staining, while yellow arrowheads in **B** indicate punctae of strong pMRLC and Stbm-Venus colocalization. Scale bars are 10 μm.

Blebbistatin, which blocks myosin motor function, was also able to cause nuclear and Pk-myc depolarization (*Figure 4G*), but with varying effectiveness. In one set of experiments, we observed Pk-myc polarization in only 2 of 18 cells scored, while in second set, we observed Pk-myc polarization in 31 of 43 cells. The DMSO-treated controls showed 34/41 and 12/13 cells with proper Pk-myc polarity, respectively. Blebbistatin-treated cells scored as having Pk-myc polarization often lacked the restricted domain of labeling at the center of the membrane seen in DMSO-treated controls, instead showing a more even distribution throughout the anterior membrane.

As expected, treatment of the embryos with either cytochalasin B or latrunculin B profoundly disrupted the actin network in notochord cells. Staining of control embryos with phalloidin shows strong cortical actin in the notochord cells (*Figure 4H*). Cytochalasin B treatment resulted in fragmentation of the actin network (*Figure 4I*), with apparent loss of the equatorial contractile ring (indicated by arrows in 4H) (*Sehring et al., 2014*), while latrunculin B resulted in a complete loss of phalloidin staining (*Figure 4J*). The embryos shown in *Figure 4H–J* were also stained for the protein atypical protein kinase C (aPKC), which is localized to the cell cortex of notochord cells, but unlike Pk and Stbm is found at both the anterior and posterior poles of the cells (*Denker et al., 2013*) (*Figure 4H′*, arrowhead). Surprisingly, a stronger disruption in the localization of aPKC was seen with cytochalasin B treatment than with latrunculin B (*Figure 4H′,J′*). We speculate that aPKC may associate with the fragmented actin filaments seen with cytochalasin treatment, resulting in it spreading across the face of the cells, while in the complete absence of filamentous actin (latrunculin B) the aPKC is not displaced and is able to remain concentrated at the center of the cell.

## Myosin remains localized in the presence of cytoskeletal inhibitors

While we observed that cytochalasin, latrunculin, and blebbistatin treatments all lead to loss of core PCP protein and nuclei localization, the same is not true for myosin localization. Using MRLC9/12-Venus as a marker, we observed that treatment of embryos with cytochalasin B or latrunculin B from

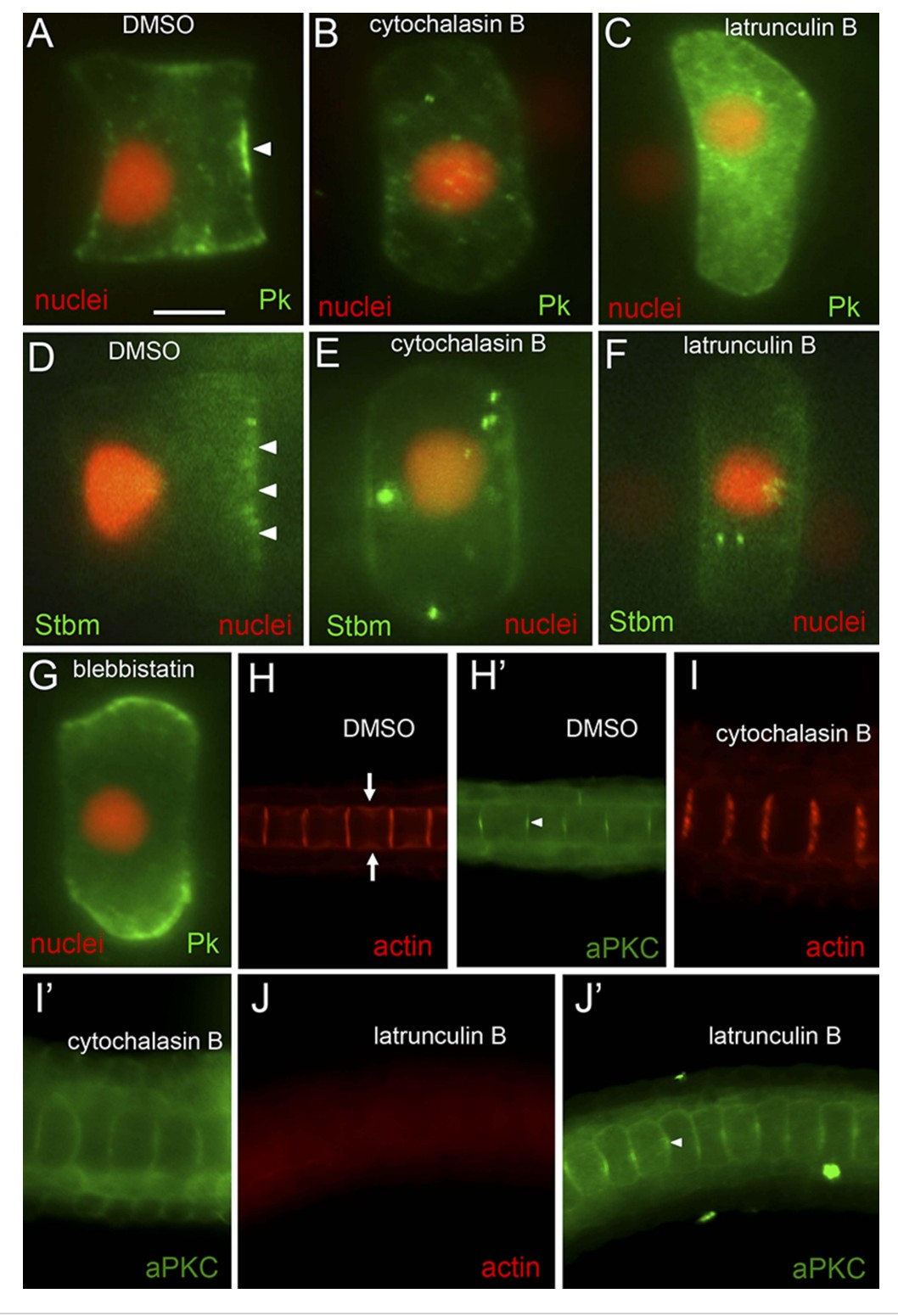

**Figure 4**. Loss of planar cell polarity protein localization following actin/myosin disruptions. (**A–G**) Single notochord cells from embryos expressing H2A-RFP (nuclear RFP), and either a Pk-myc or Stbm-Venus fusion protein (as indicated). Arrowheads indicate anteriorly localized reporter protein. Embryos were treated with either DMSO (vehicle), cytochalasin B, latrunculin B, or blebbistatin (also as indicated). (**H**, **I**, and **J**) Cortical actin staining of notochord cells with phalloidin in embryos treated with vehicle only (DMSO), cytochalasin B, or latrunculin B,
*Figure 4. continued on next page*

*Figure 4. Continued*

as indicated. Arrows in **H** indicate the equatorial contractile ring of actin. (**H′**, **I′**, and **J′**) Co-staining of samples for the cortical protein atypical protein kinase C (aPKC). Arrowhead in **H′** indicates concentrated staining at center of cell cortex. Scale bar in **A** is 5 μm.

stage 21–23 did not disrupt MRLC9/12-Venus polarization (*Figure 5A–D*), although the cells failed to extend to the same extent as the DMSO-treated controls, and the nuclei were displaced to the center of the cells. Additionally, in those cells treated with cytochalasin B, but not latrunculin B, MRLC9/12-Venus staining was punctate, albeit still anteriorly localized.

In older embryos, cytochalasin B appeared to be more disruptive. While treatment of embryos at early stage 23 to the start of stage 24 only resulted in a modest disruption of MRLC9/12-Venus polarization (*Figure 5D*), with 80% of treated cells retaining polarity vs 90% of control cells, treatment of embryos with cytochalasin B at late-stage 23 had a stronger effect, with 57% of cells scored (n = 44) having polarized MRLC9/12-Venus (vs 100% of cells in DMSO-treated embryos (n = 13); *Figure 5D*). The transition to cytochalasin B dependence from early to late-stage 23 may be tied to dramatic changes in cell architecture as the notochord begins the process of lumen formation, first visible at stage 24, a period during which the orderly A/P arrangement and polarity of cells changes dramatically (*Dong et al., 2009*).

The dependence of the anterior polarization of MRLC9/12-Venus on myosin activity was also examined using blebbistatin. Because blebbistatin inhibits myosin II Mg-ATPase activity without disassembling actin/myosin complexes (*Straight et al., 2003*), we would expect to observe a disruption in polarization of MRLC9/12-Venus only if the polarization required myosin motor activity. In embryos treated with 10 μM blebbistatin from stage 21 through stage 23, a dramatic block in cell elongation was observed, although there was no apparent disruption of MRLC9/12-Venus polarization indicating that MRLC9/12-Venus polarization does not require myosin activity (*Figure 5E,F*).

In summary, the core PCP proteins and myosins respond very differently to cytoskeletal inhibitors. The initiation and maintenance of Stbm-Venus and Pk-myc localization depends on an intact actin cytoskeleton, and addition of the myosin inhibitor blebbistatin leads to loss of Pk-myc and Stbm-Venus polarization. These same treatments also disrupted nuclei, which normally polarize posteriorly in a PCP-dependent manner (*Jiang et al., 2005*), indicating an extensive disruption to the normal polarity of notochord cells. In contrast, the anterior polarization of MRLC9/12-Venus is largely resistant to these treatments, indicating that the polarization of the myosin machinery in *Ciona* notochord cells can occur independently of the localization of the core PCP proteins Pk and Stbm.

## Dynamic repolarization upon removal of cytochalasin B

Despite the loss of A/P polarization of nuclei, Pk-myc, and Stbm-Venus in cytochalasin B-treated cells, we observed that polarity could be restored when the cytochalasin B was washed out. Within approximately 60 min of placing the embryos in cytochalasin B-free seawater, nuclei were observed to move back to the posterior membranes and resume their flattened shapes (*Figure 6A*; *Video 2*). Likewise, Pk-myc was observed to relocalize to anterior membranes upon wash out of cytochalasin B (*Figure 6B*). In assessing Pk-myc repolarization, a sample of embryos was first collected at the end of the cytochalasin B treatment to assure the efficacy of the treatment. In cells from this first group of embryos, anterior localization of Pk-myc was observed in only one of 33 cells scored. 1 hr after cytochalasin B was washed out, 13 out of 18 cells scored had re-established anterior

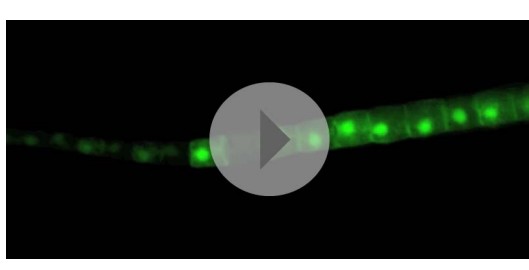

**Video 1.** Detachment of nuclei during cytochalasin B treatment. The video represents 30 min in real time, beginning approximately 10 min after the addition of cytochalasin B.

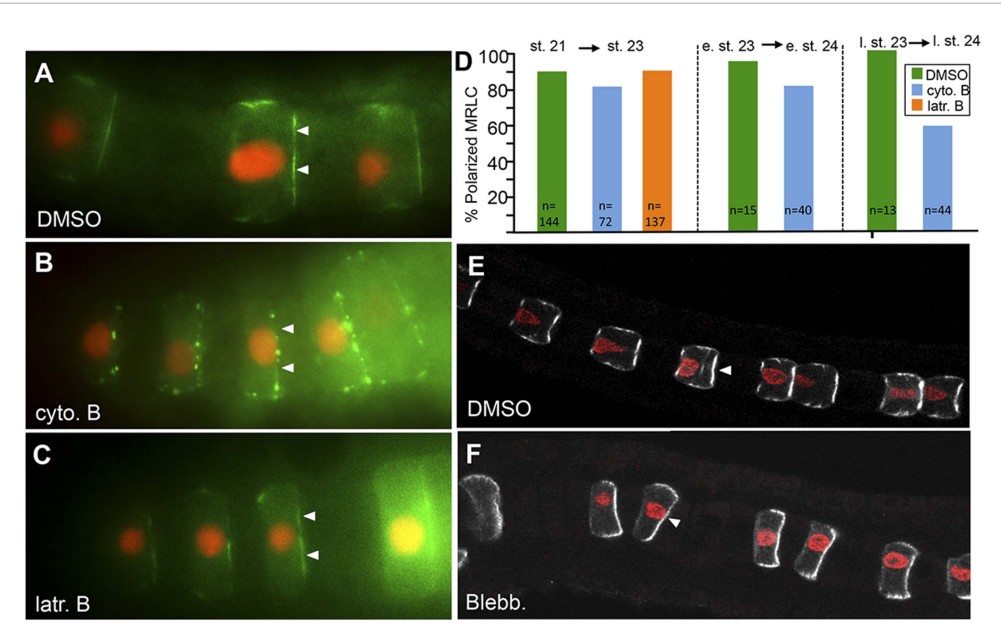

**Figure 5**. MRLC9/12 localization is resistant to cytochalasin B and blebbistatin. (**A**) Control, vehicle-treated embryo shows polarization of MRLC9/12 to the anterior membrane of notochord cells. (**B** and **C**) MRLC9/12 polarization is resistant to cytochalasin B and latrunculin B treatments. MRLC9/12-Venus is labeled green, and H2A-RFP (nuclei) is labeled red. (**D**) Percentage of notochord cells showing polarized MRLC9/12 for three treatment regimes. The number of cells scored is indicated at the bottom of the columns. e. st., early stage; l. st., late stage. (**E** and **F**) Persistence of MRLC9/12-Venus polarization (white) in embryos treated with blebbistatin or vehicle (DMSO) from stage 21 to stage 23. Arrowheads indicate anteriorly localized proteins.

localization of Pk-myc (*Figure 6B*). However, when blebbistatin was added during the recovery from cytochalasin B treatment, the ability of the cells to properly repolarize was greatly reduced (*Figure 6D*). Only 14 of 45 cells (31%) scored from 19 embryos showed proper repolarization of Pk-myc in the presence of blebbistatin vs 20 of 24 cells (83%) from 15 embryos treated with vehicle (DMSO). In contrast to cytochalasin B-treated embryos, embryos treated with latrunculin B were unable to recover either nuclear or Pk-myc polarization in the notochord, even after extensive washing.

Cytochalasin B causes extensive disruption of notochord cells. The cells undergo a dramatic shape change, the polarization of Pk-myc, Stbm-Venus, and nuclei is lost, and the localization of aPKC is disrupted. Despite this, polarization of MRLC9/12-Venus persists, and much of the actin network remains cortical, although fragmented (*Figure 4I*). Inhibition of myosin motor activity with blebbistatin gives similar effects on Pk-myc and nuclei. Thus, in blebbistatin-treated cells or cytochalasin B-treated cells, while the actin and myosin machinery are inactivated, they appear to remain polarized and cortical. Removal of embryos from cytochalasin B allows the recovery of Pk-myc and nuclei, but not in the presence of blebbistatin. Together, these results support an active role for actin/myosin machinery in repolarization of core PCP proteins and nuclei. Consistent with this, latrunculin treatment, while not disrupting MRLC9/12-Venus localization, caused a much more extensive disruption the cortical actin network than did cytochalasin B (*Figure 4J*). These cells do not recover nuclear or Pk-myc polarization when removed from latrunculin, presumably because of the more extensive loss of the actin network. In summary, despite the loss of Stbm and Pk localization in cytochalasin-treated cells a polarity persists that can restore proper core PCP and nuclei polarity. This persistent polarity appears to be the actin/myosin machinery.

## Neither establishment, maintenance nor recovery of nuclear, Pk or MRLC9/12 polarity requires the microtubule network

Previous reports in *Drosophila* have shown an important role for the microtubule network in PCP function (*Shimada et al., 2006*; *Harumoto et al., 2010*; *Olofsson et al., 2014*). The microtubule

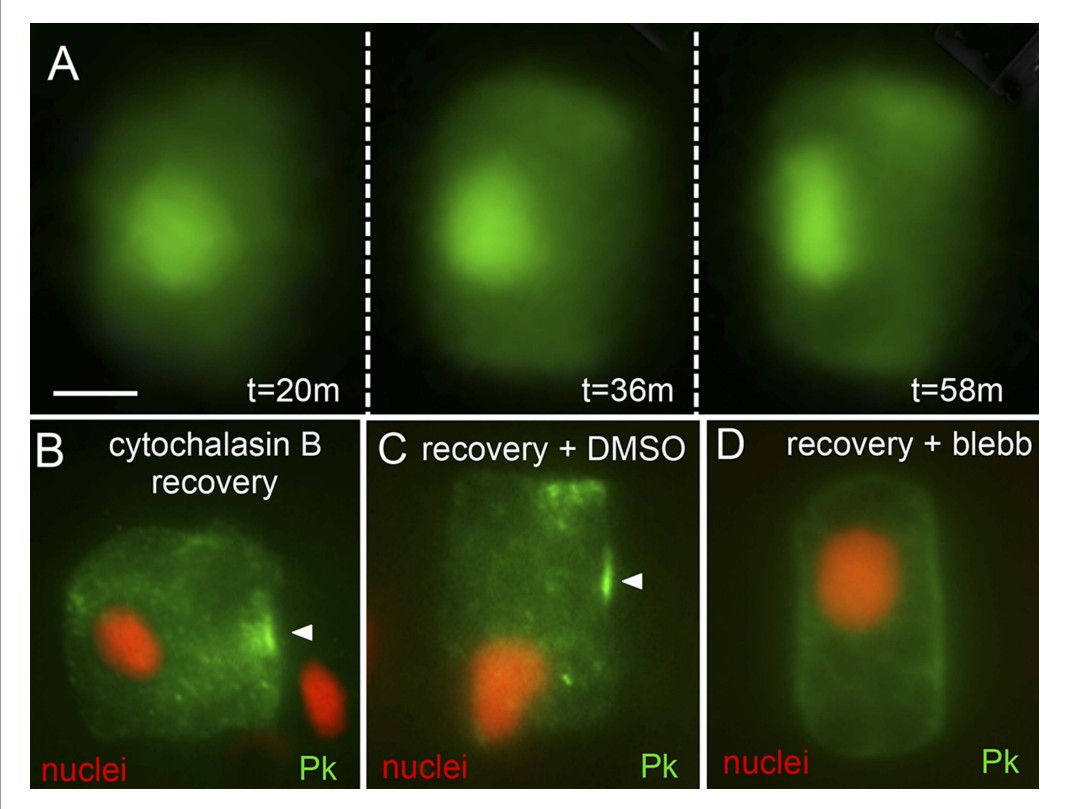

**Figure 6**. Recovery of polarity following cytochalasin B treatment. (**A**) Time lapse of nuclear repolarization following removal of embryos from cytochalasin B. Times are from end of cytochalasin B treatment. (**B**) Recovery of Pk-myc polarization following cytochalasin B treatment and wash out. Anterior is to the right in all panels. (**C** and **D**) Blebbistatin blocks recovery from cytochalsin B treatment. Following removal of embryos from cytochalsin B, embryos were either treated with vehicle (DMSO) (**C**) or 10 μm blebbistatin (**D**). Arrowheads indicate anterior localized Pk. Scale bar in **A** is 5 μm.

network in the *Ciona* notochord can be readily visualized with a GFP-labeled version of the microtubule-binding protein Ensconsin (*Figure 7A*; [*Roure et al., 2007*]), and addition of 10 μm nocodazole results in a complete disruption of the network (*Figure 7B*). To test for a role of microtubules in the establishment of polarity, embryos were treated with nocodazole at stage 21. At stage 21, notochord intercalation is complete, but no signs of A/P polarity are evident. The resulting embryos observed at stage 23 showed relatively normal notochord development with no perturbations in either nuclear position (*Figure 7C,D*) or in Pk-myc localization (*Figure 7E*). Of 26 nocodazole-treated cells examined, 21 (81%) had normal Pk-myc localization vs 23 of 25 cells (92%) for a DMSO-treated control. The addition of vinblastine, another microtubule inhibitor, at stage 21 is consistent with our nocodazole results; after vinblastine treatment at stage 21, 210 of 211 cells (99.5%) showed proper nuclear polarity, compared to 227 of 228 (99.6%) for the DMSO control (data not shown). Moreover, the maintenance of Pk-myc polarization was also not disrupted by the

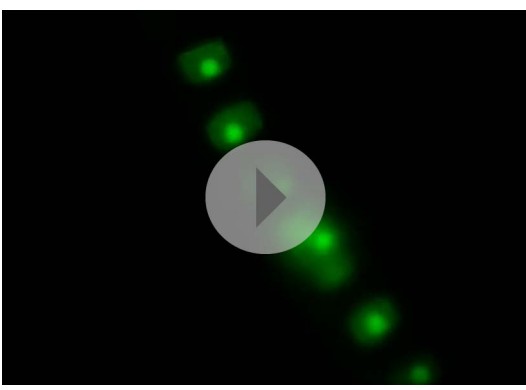

**Video 2.** Movement of nuclei to the posterior poles of notochord cells following removal in cytochalasin B. The video represents 28 min in real time, beginning approximately 10 min after removal from cytochalasin B.

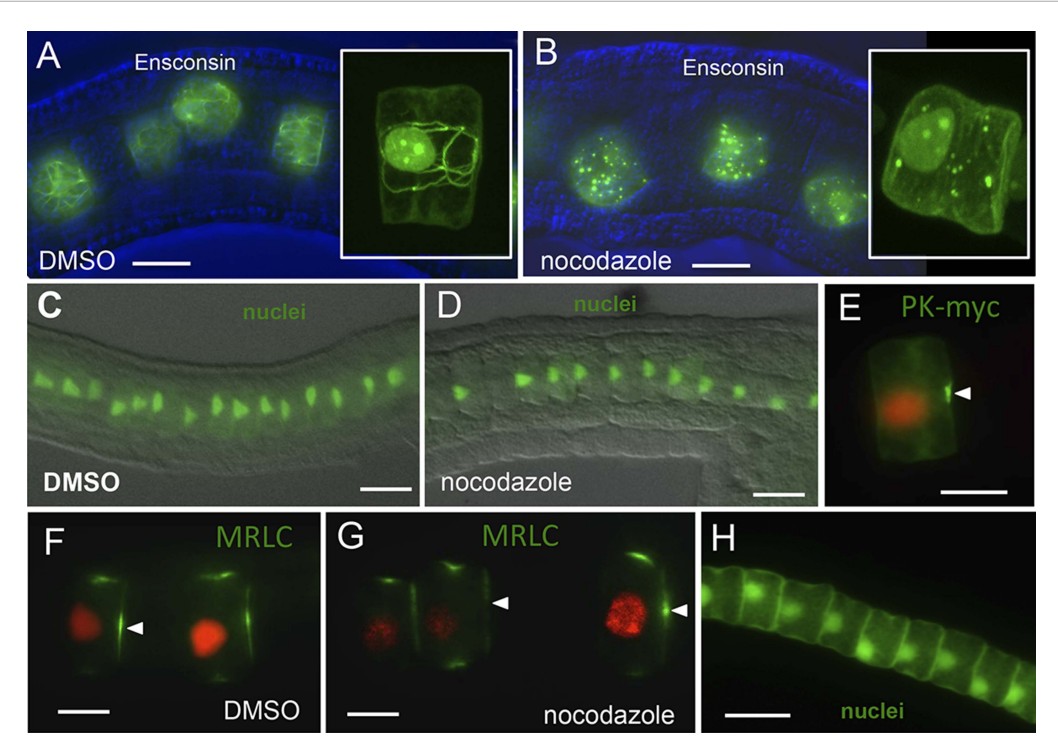

**Figure 7**. Notochord A/P polarity does not require microtubular network. (**A**) Visualization of the notochord microtubule network with GFP-tagged ensconsin, and **B**, disruption of the network with 10 µm nocodazole. (**C–E**) Nocodazole treatment from stage 21 disrupts neither nuclear (**D**) nor Pk-myc (**E**) polarity. Embryos/cells are shown at stage 23. Similarly, nocodazole treatment at stage 21 does not disrupt MRLC9/12 polarity (control, **F**; treated, **G**). (**H**) Cell nuclei properly repolarized in the presence of nocodazole following cytochalasin treatment. Arrowheads indicate anterior localized MRLC9/12. Scale bars are 10 µm.

addition of nocodazole at stage 23. In embryos treated with 10 µm nocodazole for 1 hr starting at stage 23, we observed that 42 of 45 cells (93%) scored had anterior-localized Pk-myc (vs 32 of 36 cells (89%) from DMSO-treated embryos).

Disruption of microtubules by the addition of nocodazole also had little effect on the anterior localization of MRLC9/12 in notochord cells. To test for a role in the establishment of polarity, nocodazole was added for 1 hr at late-stage 20, when notochord cell intercalation is nearly complete. In those cells scored, 90 of 102 had properly polarized MRLC9/12 (88.2%), while all DMSO-only treated notochord cells scored (111/111, 100%) showed anterior MRLC9/12 (*Figure 7F,G*). When treated later, at stages 22–23 (well after intercalation is complete) polarization of MRLC9/12-Venus was nearly unaffected in nocodazole-treated embryos; 166 of 167 (99%) of notochord cells counted showed anterior MRLC. Control animals showed localization in 100% of cells observed (61/61). Finally, addition of nocodazole did not disrupt recovery of polarity following cytochalasin B treatment. We observed that nuclei were able to properly repolarize (65/66, 98.5%) following cytochalasin wash out in the presence of 10 µm nocodazole. (*Figure 7H*).

Our results do not support a role for microtubules in the polarization of nuclei, Pk, or MRLC9/12. When nocodazole is added at a developmental stage immediately preceding that in which polarization of Pk-myc and MRLC9/12 is first evident, very little disruption of the polarity of these molecules is observed later. It is unknown what mechanism leads to the polarization of these molecules between stages 21 and 22, and an earlier, but yet unknown, polarity cue may be present. We have observed that the addition of nocodazole earlier interferes with intercalation, and that the onset of A/P polarization and intercalation appear to be linked, leaving open the possibility of an earlier role of microtubules in A/P polarity. Nevertheless, the role of microtubules in the localization of core PCP proteins in the *Ciona* notochord differs significantly from the localization

of these proteins in other models for PCP (*Shimada et al., 2006*; *Harumoto et al., 2010*; *Olofsson et al., 2014*).

## Complex localization of Myosin10/11/14/9 and MRLC9/12 in the absence of PCP signaling

The functional link between myosin polarization and the core PCP pathway was further investigated by examining Myosin10/11/14/9-myc and MRLC9/12-Venus localization in the *C. savignyi aim* line, which carries a null mutation in the *pk* gene (*Jiang et al., 2005*). Despite the absence of Pk activity, a few notochord cells in *aim* embryos are still able to intercalate. These notochord cells, which are usually but not exclusively in the posterior notochord, individually display a polarity, as is evident by the localization of the nuclei to either the anterior or posterior poles of intercalated cells. However, the polarity is not coordinated between cells, and they show no bias with respect to the A/P axis (*Kourakis et al., 2014*).

We assayed the localization of Myosin10/11/14/9-myc or MRLC9/12-Venus in 34 cells from 23 *aim* embryos, 25 having posterior nuclei and 9 having anterior nuclei (*Figure 8A*). We observed two distinct phenotypes for myosin localization. For those cells with posterior nuclei (i.e., the same as observed in wild-type embryos), we observed that both Myosin10/11/14/9-myc and MRLC9/12-Venus were polarized anteriorly (*Figure 8B,D*) as normal. By contrast, for those cells with anterior nuclei, we were unable to detect either posterior or anterior enrichment of myosin (with one exception), *Figure 8A,C,E*.

Thus, despite disruption of core PCP protein localization using cytoskeletal inhibitors, or by loss of Pk activity as is the case with the *aim* mutant, myosin can show a wild-type polarization pattern. Nevertheless, we do observe a requirement for Pk activity in properly polarizing Myosin10/11/14/9-myc and MRLC9/12-Venus in a subset of notochord cells in aim embryos.

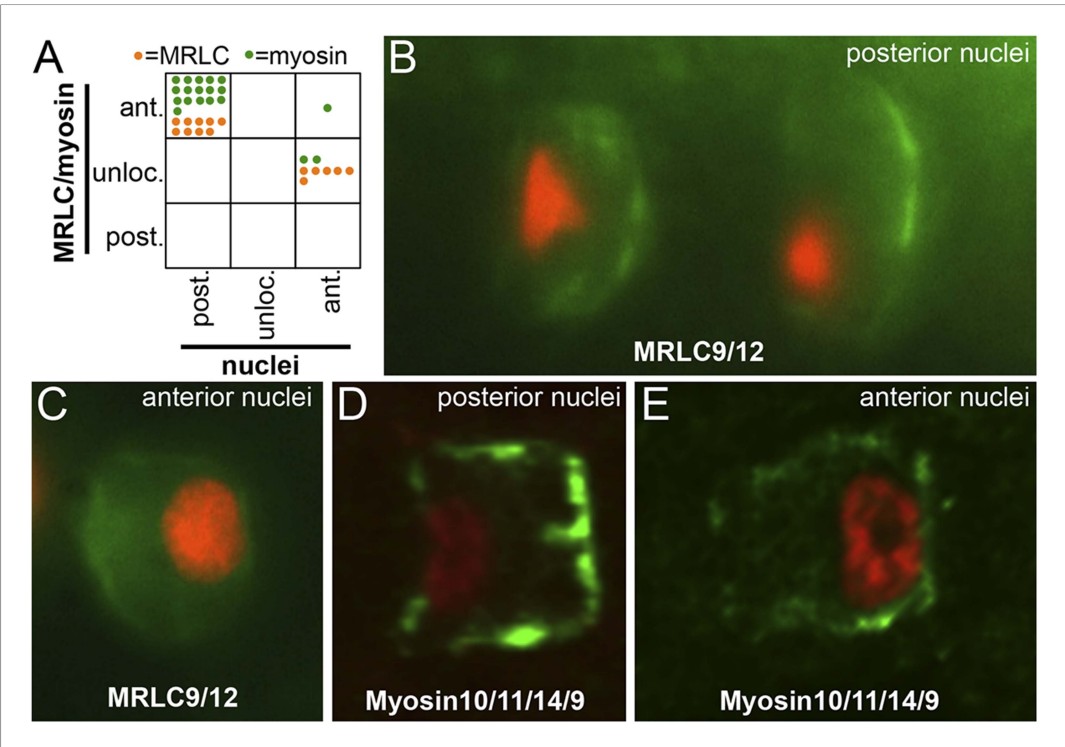

**Figure 8**. Myosin10/11/14/9 and MRLC9/12 localization in homozygous *aim* embryos. (**A**) Distribution of nuclear and Myosin10/11/14/9 (green) or MRLC9/12 (orange) localization phenotypes (ant., anterior; post., posterior; unloc., unlocalized) in *aimless* embryos. Each dot represents a single scored cell. (**B** and **C**) MRLC9/12-Venus (green) localization in cells with posterior and anterior nuclei (red), respectively. (**D** and **E**) Myosin10/11/14/9-myc (green) localization in cells with posterior and anterior nuclei (red), respectively. Anterior is to the right in all panels.

## Discussion

The results presented here demonstrate a dynamic interaction between PCP components and the actin/myosin cytoskeleton in establishing and maintaining notochord cell AP polarity. A linear pathway from the activation of the core PCP components Fz/Dsh to polarization of the actin cytoskeleton via Daam1 and RhoA has been well described, as reviewed by *Wallingford (2012)*. PCP-directed polarization of actin is also seen in the *Drosophila* wing, where the actin-rich prehair is targeted to the distal side, opposite where pk and stbm are enriched (*Wong and Adler, 1993*). Our results in the *Ciona* notochord indicate a more complex interaction in which these components are mutually reinforcing rather than being in a simple linear pathway. Additionally, and in contrast to results in other experimental systems (*Shimada et al., 2006*; *Harumoto et al., 2010*; *Vladar et al., 2012*; *Matis et al., 2014*), we find no evidence of a role for the microtubule network in the localization of either PCP components Pk and Stbm, or MRLC9/12.

We report here for the first time that myosin machinery (Myosin10/11/14/9 and MRLC9/12) is anteriorly polarized in *C. intestinalis* notochord cells and shows extensive colocalization with the core PCP protein Stbm, suggesting a functional interaction between them. This is further supported by the fact that the disruption of the actin/myosin cytoskeleton with cytochalasin, latrunculin, or blebbistatin all result in the loss of polarization of Pk and Stbm. Conversely, these same treatments leave the polarization of MRLC9/12 largely intact. Thus, Myosin10/11/14/9 in the notochord appears to localize in the absence of core PCP protein localization, similar to the situation for the atypical myosin Dachs in *Drosophila*, which is also polarized independently of the core PCP pathway (*Brittle et al., 2012*). It should be noted, however, that no direct ortholog of Dachs is found in chordate genomes (*Mao et al., 2006*). Moreover, the persistence of MRLC9/12 localization when Pk and Stbm are delocalized (e.g., cytochalasin treatment), and the requirement for myosin activity during re-polarization after cytochalasin treatment, suggest that myosin polarization in *Ciona* notochord cells could act upstream of core PCP localization. However, it is important to note that in the developing notochord, Pk, Myosin10/11/14/9, and MRLC9/12 all appear to simultaneously localize in the A/P axis between stages 21 and 22 (*Figure 2* and [*Kourakis et al., 2014*]). Thus, the timing of events would argue against a simple model in which myosin localization precedes and directs Pk polarization. Moreover, it has also been observed that polarization of Stbm lags behind that of Pk (and thus also of myosin) (*Jiang et al., 2005*), indicating that the core PCP machinery is still assembling at the time we first see polarized myosin.

The mechanism and identity of the polarizing cue that leads to the simultaneous polarization of Pk, Myosin10/11/14/9, and MRLC9/12 remains mysterious. The observations presented here in a Pk null mutant line (*aim*) provides clues to the nature of this signal. Myosin polarization was disrupted in approximately one third of the notochord cells scored in *aim* embryos. While the *pk* allele in *aim* contains an ∼200 base-pair deletion that spans one intron and an adjacent exon, creating a premature stop codon and a likely null allele (*Jiang et al., 2005*), and the *Ciona* genome contains only a single *pk* gene, one possible explanation for the variability between cells is that residual Pk-independent PCP activity remains in a subset of the notochord cells in *aim* embryos. However, by other measures of cell polarity, such as extension and protrusive activity along the mediolateral axis during intercalation, and the membrane localization of Dishevelled, we see no evidence for residual PCP activity in *aim* embryos, and moreover, all notochord cells appear to be uniformly disrupted (*Jiang et al., 2005*; *Veeman et al., 2008*; *Kourakis et al., 2014*). Thus, we surmise that many cells in the notochord are able to properly polarize myosin without input from the core PCP pathway. The notochord cells in *aim* embryos show a highly consistent pattern in which those having posterior localized nuclei (i.e., the wild-type pattern) have anterior polarized myosin, while those with inverted nuclei polarity (anterior) lacked detectable myosin polarity, rather than having inverted myosin polarity. We speculate that those cells in *aim* embryos showing wild-type nuclear polarity are correctly aligned with a global polarizing cue, and can polarize their myosin complex, while those cells showing anterior nuclei are not in alignment, and myosin does not polarize properly. The role of the core PCP pathway, which is lacking in the *aim* cells, is thus to act locally to bring the cells into proper alignment. Despite this observation, disruption of Pk and Stbm localization with cytochalasin C, latrunculin, or blebbistatin has relatively little effect on MRLC9/12 localization. One potentially important difference between these two examples is that the *aim* mutants are null for *pk* from fertilization, while the cytoskeletal inhibitors are added post-intercalation. Thus, while we see no evidence of A/P polarization until approximately

1 hr after the completion of intercalation (e.g., *Figure 2*), earlier events during intercalation may be required for the establishment of A/P polarity. The nature of these events and the mechanism that switches notochord polarity from mediolateral to A/P remain unknown.

## Materials and methods

### Animal culture

Adult *C. intestinalis* were collected from Santa Barbara Harbor or purchased from M-REP (Carlsbad, CA, USA) and kept at a facility supplied with raw seawater at the University of California, Santa Barbara. Gametes were isolated, mixed, and dechorionated as previously described. Anatomical landmarks were used to stage animal development (*Hotta et al., 2007*).

### Transgenesis

Plasmid constructs were electroporated into one-cell stage embryos (*Christiaen et al., 2009*). The electroporation procedure typically results in mosaic expression, allowing us to score expressing cells in a background of non-expressing neighbors. Nuclei were visualized either using an existing *C. intestinalis* stable transgenic line expressing GFP under control of the brachyury promoter (*Joly et al., 2007*), or by electroporation with Bra > GFP::H2A::RFP (*Kourakis et al., 2014*).

### Cytoskeletal inhibitors

Cytochalasin B (C2743, Sigma, St. Louis, MO) was provided at 10 mg/ml in DMSO and diluted in seawater to 10 µg/ml. Blebbistatin (B0560, Sigma) was prepared as 10 mM in DMSO and diluted in seawater to a final concentration of 10 µM. Nocodazole (M1404, Sigma) was diluted to 10 mM in DMSO and used at a final concentration of 10 µM in seawater. Latrunculin B (L5288, Sigma) was prepared as a 10 mg/ml stock solution and diluted to a working concentration of 2 µg/ml or 10 µg/ml in sea water; both concentrations yielded similar phenotypes. Vinblastine (V1377, Sigma) was added to a final concentration of (6 µg/ml). For each of these drugs, DMSO alone in seawater served as a negative control. An additional positive control was performed for working concentrations of nocodazole; for any trial using the microtubule inhibitor, we also confirmed that exposure to it resulted in cleavage arrest in embryos.

### Plasmid constructs

A *C. intestinalis* MRLC cDNA (KH2012:KH.C11.143; [*Satou et al., 2005*]) was PCR amplified (forward primer: ATGTCGAGCCGACGAACTAAAAA; reverse primer: AATGTCATCTTTTTCTTTAGCGCCAT) and cloned into pDONR221. This was recombined with the *Brachyury* promoter (forward primer: TAACGACGATTGTTCCGTCA; reverse primer: TATAGGTTTGTAACTCGCACTGAGC) donor construct into pSP72-R3-ccdBCmR-R5-Rfa-Venus (*Roure et al., 2007*) to generate *Bra > MRLC-Venus*.

The *C. intestinalis* Stbm cDNA (KH2012:kh.C4.173 [*Satou et al., 2005*]) generated previously (*Jiang et al., 2005*) was cloned into pDONR221, and recombined with the *Brachyury* promoter donor construct (above) into pSP72-R3-ccdBCmR-R5-Venus-Rfa (*Roure et al., 2007*) to generate *Bra > Venus-Stbm*.

The *Bra* promoter sequence was cloned into pSP72BSSPE-SwaI-Rfa and recombined with an entry vector encoding amino acids 18–283 of human ensconsin (*pEntr > Ensconsin-3xGFP*) (*Roure et al., 2007*) to generate *Bra > Ensconsin-3xGFP*.

A gBlock fragment (IDT, Coralville, IA) containing approximately 1 kb of the C-terminal portion of Myosin 10/11/14/9 (Kh. C11.456; [*Satou et al., 2005*]) cDNA was generated to include a myc tag (GCATCAATGCAGAAGCTGATCTCAGAGGAGGACCTG) inserted immediately 5′ of the stop codon. This fragment was cloned into the full-ORF clone of Myosin 10/11/14/9 (cima8817) at the SacII and Nde1 sites. This results in a full-length Myosin 10/11/14/9 cDNA with a C-terminal myc tag, including 5′ and 3′ UTRs. This donor construct was recombined with a *Bra* promoter containing destination vector to generate *Bra > Myosin 10/11/14/9-myc*.

### Immunohistochemistry

Embryos were fixed in 2% paraformaldehyde in seawater for 1 hr then washed 4 times in PBST, 10 min each. Embryos were incubated in PBST + 5% goat serum for 1 hr at room temperature, or overnight at 4°C, with the primary antibody diluted 1:1000 (anti-GFP, anti-myc, anti-RFP; Invitrogen, Grand Island, NY), 1:650 (anti-HA; Millipore, Billerica, MA), 1:300 (anti-PKC ζ; Santa Cruz Biotechnology,

Dallas, TX), or 1:250 (anti-phospho-Myosin Light Chain 2 [Ser19]; Cell Signaling Technology, Danvers, MA). Animals were washed 5× for 10 min in PBST and then placed in an appropriate secondary antibody, and incubated as described for the primary antibody. Secondary antibodies used were anti-mouse or anti-rabbit Alexa Fluor-labeled antibodies (Invitrogen) with a range of excitation/emission spectra, depending on the experiment. Samples were washed 4 to 10× in PBST following secondary incubation. For microscopy, labeled embryos were immobilized onto cover slips coated with 0.08% Poly-L-lysine. Fixed embryos were cleared in 80% glycerol or in an isopropyl alcohol series followed by 2:1 benzyl alcohol:benzoyl benzoate (BABB).

## Injection of *aimless* embryos

Embryos were collected from natural spawning of *aimless* mutants, dechorionated, and injected as described previously (*Kourakis et al., 2014*). An injection solution containing 0.5 µg of Myosin10/11/14/9-myc or MRLC9/12-Venus, 0.25 µg of Bra > H2A::RFP, and 0.5 mg/ml Fast Green (Sigma) was injected into one cell of a two or four-cell embryo. Embryos were fixed and immunostained (as described above) when they reached stage 23.

## Acknowledgements

This work was supported by award GM088997 from the National Institutes of Health.

## Additional information

### Funding

| Funder | Grant reference | Author |
|---|---|---|
| National Institutes of Health | GM088997 | Erin Newman-Smith, Matthew J Kourakis, Wendy Reeves, Michael Veeman, William C Smith |

The funder had no role in study design, data collection and interpretation, or the decision to submit the work for publication.

### Author contributions

EN-S, MJK, WR, MV, WCS, Conception and design, Acquisition of data, Analysis and interpretation of data, Drafting or revising the article

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
