## [Decision Letter]

Thank you for sending your work entitled “Reciprocal and dynamic polarization of PCP core components and myosin machinery” for consideration at *eLife*. Your article has been favorably evaluated by K VijayRaghavan (Senior editor and Reviewing editor) and two reviewers, one of whom, Suzanne Eaton, has agreed to share her identity.

The Senior editor and the reviewers discussed their comments before we reached this decision, and the editor has assembled the following comments to help you prepare a revised submission.

Newman-Smith et al. have investigated how the core planar cell polarity (PCP) pathway coordinates cell polarization in the *Ciona* notochord. This is a particularly interesting one in which to study PCP. Previous work from the same group has shown that polarity of Core PCP proteins change dynamically as cells shift from intercalation to AP polarization, so some new insights into the diversity of polarization signals may emerge from this system. Furthermore the Core PCP proteins appear to be wired up somewhat differently than in *Drosophila* (Dsh and Pk are on the same side of the cell). Here, they have demonstrated that myosin, like Pk, becomes anteriorly polarized, and they present data suggesting that the actin-myosin cytoskeleton is important to maintain anterior localization of Pk and Stbm. This requirement is bidirectional to some extent because myosin localization is occasionally perturbed in the Pk mutant *aimless*. This would be a novel contribution to the PCP field: while it has been clear that Core PCP can influence the actin cytoskeleton, we are not aware of any data showing the reverse.

There are many important issues that need to be addressed before the paper is suitable for publication. A principal concern is the specificity of the phenotypes caused by perturbing actin and the myosin components MRLC9/12 and Myosin10/11/14/9. In general, the study relies too extensively on drug treatments without addressing the obvious caveat that these could exert non-specific effects by acting on neighboring tissues. For example, knock-down of the MLRC9/12 and/or Myosin10/11/14/9 genes and analyses of the effects on Pk and Stbm polarization would adequately complement the blebbistatin study. Much value will be added by independent ways of down-regulating cytoskeletal components, such as the use of Morpholinos or more recent CRISPR methods to knockdown components specifically in the notochord. If the authors can manage these additional experiments in a reasonable time-frame, the results and analysis could place the work on a more robust footing.

Further, how dramatically is the cell cortex affected? The authors have shown that Pk and Stbm localization is perturbed, but are other cortical proteins insensitive to these treatments? Drug treatments that depolymerize the actin cytoskeleton are shown to disrupt the anterior localization of Pk-myc and Stbm-Venus fusion proteins and the posterior localization of the nuclei, but not the anterior localization of MLRC9/12 and Myosin10/11/14/9 fusion proteins. The authors show that polarity can be recovered upon washing the drug cytochalasin, but not latrunculin. Blebbistatin, an inhibitor of myosin II, is presented as blocking the recovery of Pk-myc and Stbm-Venus repolarization following cytochalasin washout. As shown this data is not convincing. Blebbistatin does not seem to disrupt myosin localization. Is there something that we are missing?

There are several other important issues that also need addressing. In general, the results need to be presented with clear interpretation or conclusions about the biological mechanisms which they reveal. The absence of this is pervasive from the Abstract through the Results and Discussion. The discussion invokes elusive and hypothetical cell autonomous and global polarization mechanisms, which weaken the paper rather than clarifying the main conclusions. The final model summarizes the observations while it could instead propose a mechanism to explain how PCP components and myosin regulate each other's localization. The authors use the drug nocodazole to effectively disrupt the microtubule cytoskeleton and observed no effect on the polarization of the PCP components. They conclude that the microtubules are dispensable for polarized distribution of Pk and Stbm. However, these treatments were apparently performed at a stage following the initial polarization, meaning that microtubules may be dispensable for the local steady-state maintenance of polarized distribution but not for the initial polarization. It would be important to perform the treatments and analyses at different time points, especially before the initial polarization, to conclude on this negative result. For myosin, only subcellular localization is showed while one could use anti phospho-myosin antibodies (as in Sherrard et al, Curr Biol, 2010, 20(17), 1-12) to ask if active myosin is localized to the anterior side of the cells. Colocalization analyses between PCP and Myosin components in both control and drug-treated (or knockdown) embryos would probably illuminate specific aspects of the crosstalk. It would be interesting to test if PCP components and myosin colocalize in control embryos whereas this would be disrupted upon cytochalasin, latrunculin and/or blebbistatin treatments.

Other comments:

1) Section headed “Dynamic Repolarization upon Removal of Cytochalasin B”. The data showing the delocalized cortical distribution of Pk in blebbistatin-treated animals should be shown and quantified because the effects of blebbistatin are central to the model.

2) Section headed “Myosin is Anteriorly Polarized in Notochord Cells”. The authors describe the localization of Myosin light and heavy chain as being more dispersed. Looking at the figure, the distribution of myosin heavy chain actually seems quite focused. Interestingly, it looks as if myosin heavy chain localization extends further into the cell and isn't just cortical, but it is hard to tell how different it is from MRLC because the images are somewhat over-exposed. It would be nice to see shorter exposures and some discussion of the differences in localization if they are there. Is myosin covering the nuclei?

3) Section “Myosin Remains Localized in the Presence of Cytoskeletal Inhibitors” (last paragraph). It would be good to see the pictures of the nocodozole experiments, rather than just the quantifications in the text.

4) Section entitled “Complex Localization of Myosin10/11/14/9 and MRLC9/12 in the Absence of PCP Signaling” and Discussion. In their model, the authors treat *aimless* mutants as null for Core PCP in general but this may not be so. It could be possible that residual activity of Core PCP proteins is responsible for the incomplete disruption of myosin polarity. Is polarization of Stbm abrogated in *aimless* mutants?

---

## [Author Response]

*There are many important issues that need to be addressed before the paper is suitable for publication. A principal concern is the specificity of the phenotypes caused by perturbing actin and the myosin components MRLC9/12 and Myosin10/11/14/9. In general, the study relies too extensively on drug treatments without addressing the obvious caveat that these could exert non-specific effects by acting on neighboring tissues. For example, knock-down of the MLRC9/12 and/or Myosin10/11/14/9 genes and analyses of the effects on Pk and Stbm polarization would adequately complement the blebbistatin study. Much value will be added by independent ways of down-regulating cytoskeletal components, such as the use of Morpholinos or more recent CRISPR methods to knockdown components specifically in the notochord. If the authors can manage these additional experiments in a reasonable time-frame, the results and analysis could place the work on a more robust footing*.

We agree that knockout or knockdown of MRLC9/12 and/or Myosin10/11/14/9 would be informative, but because of their early expression, these molecules will likely also be involved in notochord intercalation. Therefore we would not be able to assess their role in A/P polarity because the morphogenesis would arrest too early in development. This is exactly what we see when cytoskeletal disrupting drugs are added during intercalation. All existing protocols for gene knockdown in *Ciona* require introduction of morpholinos or CRISPR plasmids at the one cell stage, and methods that would allow us to temporally target the knockdown postintercalation have not been developed in *Ciona*. Moreover, we feel that the reviewers’ concern about “non-specific effects by acting on neighboring tissues” should not be a major concern. These are the conclusions we are drawing from the drug studies:

• Myosin is still polarized despite Pk and Stbm (and nuclei) being delocalized by the drugs. Therefore, there is not a strict requirement for core PCP polarization for myosin polarization.

• Repolarization following cytochalasin wash-out is blocked by blebbistatin. Therefore, myosin activity can localize pk.

• While the drugs we are using will disrupt cytoskeleton embryo-wide, in *Ciona* the core PCP proteins are only expressed in the notochord, and the only known phenotypes from loss of PCP activity are in the notochord (our published data).

Despite this, we attempted CRISPR knockout for both MRLC9/12 and Myosin10/11/14/9 using a recently published protocol for *Ciona*. We’ve tried two guide RNAs for each target, and so far we have been unable to detect any evidence of gene editing using either nuclease assays or direct sequencing (but we have had success targeting other genes). We will continue to try targeting MRLC9/12 and Myosin10/11/14/9 by CRISPR, but we do not feel that we can trouble-shoot this approach within the time-frame of the revision, and we point out that the reviewers requested these experiments “if the authors can manage these additional experiments in a reasonable time-frame”. We will continue to work on the MRLC9/12 and Myosin10/11/14/9 knockdown, but this may require development of tools that allow inducible expression.

*Further*, *how dramatically is the cell cortex affected? The authors have shown that Pk and Stbm localization is perturbed, but are other cortical proteins insensitive to these treatments?*

We’ve looked at the localization of aPKC, another protein localized to the cortex and A/P polarized (and one that we can easily detect with an antibody). We find that its localization is also disrupted by cytochalasin (see Figure 4). In the same figure we also present data on cortical actin.

*Drug treatments that depolymerize the actin cytoskeleton are shown to disrupt the anterior localization of Pk-myc and Stbm-Venus fusion proteins and the posterior localization of the nuclei, but not the anterior localization of MLRC9/12 and Myosin10/11/14/9 fusion proteins. The authors show that polarity can be recovered upon washing the drug cytochalasin, but not latrunculin. Blebbistatin, an inhibitor of myosin II, is presented as blocking the recovery of Pk-myc and Stbm-Venus repolarization following cytochalasin washout. As shown this data is not convincing*. *Blebbistatin does not seem to disrupt myosin localization. Is there something that we are missing?*

Based on published work ([38], Science), we would not expect to see disruption in myosin localization. Blebbistin inhibits myosin II Mg-ATPase activity without disassembling the actin/myosin complex. We have modified the text to make this clear.

*There are several other important issues that also need addressing. In general, the results need to be presented with clear interpretation or conclusions about the biological mechanisms which they reveal. The absence of this is pervasive from the Abstract through the Results and Discussion. The discussion invokes elusive and hypothetical cell autonomous and global polarization mechanisms, which weaken the paper rather than clarifying the main conclusions. The final model summarizes the observations while it could instead propose a mechanism to explain how PCP components and myosin regulate each other's localization*.

We have extensively re-written the text according to the reviewers’ suggestion to make it less speculative. Accordingly, we have removed the cartoon that summarizes our speculation.

*The authors use the drug nocodazole to effectively disrupt the microtubule cytoskeleton and observed no effect on the polarization of the PCP components. They conclude that the microtubules are dispensable for polarized distribution of Pk and Stbm. However, these treatments were apparently performed at a stage following the initial polarization, meaning that microtubules may be dispensable for the local steady-state maintenance of polarized distribution but not for the initial polarization. It would be important to perform the treatments and analyses at different time points, especially before the initial polarization, to conclude on this negative result*.

We now include new data in which nocodazole was added at stage 21, immediately after intercalation, and before we observe localized Pk or myosin (Figure 7). We see no significant disruption in polarity. Thus we’ve now tested for an effect on nocodazole on the establishment of polarity, the disruption of polarity once it is established, and the recovery of polarity following cytochalasin wash-out.

*For myosin, only subcellular localization is showed while one could use anti phospho-myosin antibodies (as in Sherrard et al, Curr Biol, 2010, 20(17), 1-12) to ask if active myosin is localized to the anterior side of the cells. Colocalization analyses between PCP and Myosin components in both control and drug-treated (or knockdown) embryos would probably illuminate specific aspects of the crosstalk. It would be interesting to test if PCP components and myosin colocalize in control embryos whereas this would be disrupted upon cytochalasin, latrunculin and/or blebbistatin treatments*.

We have done the suggested co-staining for phospho-myosin and the core-PCP proteins Pk and Stbm. Interestingly, we see stronger co-localization of Stbm with p-myosin than we do with Pk. These results are now presented in Figure 3.

The suggestion that we examine Stbm, Pk and phosphomyosin overlap following treatment with cytoskeletal inhibiting drugs is not feasible. The disruption in the localization of Pk and Stbm following treatment disperses the protein to the degree that in most cases make them undetectable above the level of background fluorescence.

*Other comments*:

*1) Section headed “Dynamic Repolarization upon Removal of Cytochalasin B”. The data showing the delocalized cortical distribution of Pk in blebbistatin-treated animals should be shown and quantified because the effects of blebbistatin are central to the model*.

This is now included.

2) Section headed “Myosin is Anteriorly Polarized in Notochord Cells”. The authors describe the localization of Myosin light and heavy chain as being more dispersed. Looking at the figure, the distribution of myosin heavy chain actually seems quite focused. Interestingly, it looks as if myosin heavy chain localization extends further into the cell and isn't just cortical, but it is hard to tell how different it is from MRLC because the images are somewhat over-exposed. It would be nice to see shorter exposures and some discussion of the differences in localization if they are there. Is myosin covering the nuclei?

It would be difficult to conclude too much because we are looking at exogenous proteins, and the differences may be due to differences in expression levels. Our new anti-phosphomyosin data show endogenous protein (Figure 3); and in this case, the protein looks very cortical. We have replaced the myosin heavy chain with an image from a cell expressing a lower amount of exogenous myosin, on the assumption that this more closely matches the endogenous protein.

*3) Section “Myosin Remains Localized in the Presence of Cytoskeletal Inhibitors” (last paragraph). It would be good to see the pictures of the nocodazole experiments, rather than just the quantifications in the text*.

We have included these pictures in Figure 7.

*4) Section headed “Complex Localization of Myosin10/11/14/9 and MRLC9/12 in the Absence of PCP Signaling” and Discussion. In their model, the authors treat aimless mutants as null for Core PCP in general but this may not be so. It could be possible that residual activity of Core PCP proteins is responsible for the incomplete disruption of myosin polarity. Is polarization of Stbm abrogated in* aimless *mutants?*

We modified the manuscript to include this possibility, be we think this is unlikely. We’ve looked at a number of cell polarity measurements in the *aimless* mutant, and in all cases our results are consistent with a complete loss of core PCP activity. While we haven’t looked at Stbm in *aimless*, we have looked at Dishevelled, and it’s completely delocalized. We’ve added the follow section to the Discussion section:

“While the *pk* allele in *aim* contains an ∼200 base-pair deletion that spans one intron and an adjacent exon […] all notochord cells appear to be uniformly disrupted (19; 22; 41).”